# Antibacterial Effects of Bicarbonate in Media Modified to Mimic Cystic Fibrosis Sputum

**DOI:** 10.3390/ijms21228614

**Published:** 2020-11-16

**Authors:** Pongsiri Jaikumpun, Kasidid Ruksakiet, Balázs Stercz, Éva Pállinger, Martin Steward, Zsolt Lohinai, Orsolya Dobay, Ákos Zsembery

**Affiliations:** 1Department of Oral Biology, Semmelweis University, H-1089 Budapest, Hungary; pongsirij@nu.ac.th (P.J.); ksd13rsk@gmail.com (K.R.); martin.steward@manchester.ac.uk (M.S.); 2Department of Conservative Dentistry, Semmelweis University, H-1088 Budapest, Hungary; lohinai.zsolt@dent.semmelweis-univ.hu; 3Institute of Medical Microbiology, Semmelweis University, H-1089 Budapest, Hungary; stercz.balazs@med.semmelweis-univ.hu (B.S.); dobay.orsolya@med.semmelweis-univ.hu (O.D.); 4Institute of Genetics, Cell, and Immunobiology, Semmelweis University, H-1089 Budapest, Hungary; eva.pallinger@gmail.com; 5School of Medical Sciences, University of Manchester, Manchester M13 9PL, UK

**Keywords:** cystic fibrosis, chronic inflammation, bacteria, biofilm, mucus, bicarbonate, pH

## Abstract

Cystic fibrosis (CF) is a hereditary disease caused by mutations in the gene encoding an epithelial anion channel. In CF, Cl^−^ and HCO_3_^−^ hyposecretion, together with mucin hypersecretion, leads to airway dehydration and production of viscous mucus. This habitat is ideal for colonization by pathogenic bacteria. We have recently demonstrated that HCO_3_^−^ inhibits the growth and biofilm formation of *Pseudomonas aeruginosa* and *Staphylococcus aureus* when tested in laboratory culture media. Using the same bacteria our aim was to investigate the effects of HCO_3_^−^ in artificial sputum medium (ASM), whose composition resembles CF mucus. Control ASM containing no NaHCO_3_ was incubated in ambient air (pH 7.4 or 8.0). ASM containing NaHCO_3_ (25 and 100 mM) was incubated in 5% CO_2_ (pH 7.4 and 8.0, respectively). Viable *P. aeruginosa* and *S. aureus* cells were counted by colony-forming unit assay and flow cytometry after 6 h and 17 h of incubation. Biofilm formation was assessed after 48 h. The data show that HCO_3_^−^ significantly decreased viable cell counts and biofilm formation in a concentration-dependent manner. These effects were due neither to extracellular alkalinization nor to altered osmolarity. These results show that HCO_3_^−^ exerts direct antibacterial and antibiofilm effects on prevalent CF bacteria.

## 1. Introduction

Cystic fibrosis (CF) is a genetic disease caused by mutations in the gene encoding cystic fibrosis transmembrane conductance regulator (CFTR) protein [1]. CFTR is an epithelial anion channel that primarily mediates chloride and bicarbonate ion secretion [2,3]. In some epithelia, it may also regulate sodium absorption via the epithelial sodium channel (ENaC) [4]. These functions are essential for regulating fluid movement across multiple epithelial surfaces in the body. Therefore, mutations in this channel protein result in multiple-organ dysfunction, which profoundly affects the life expectancy of CF patients [5].

Chronic lung disease is the leading cause of death in CF patients. In the CF lung, CFTR dysfunction causes a decrease in Cl^−^ and HCO_3_^−^ secretion, and an increase in Na^+^ absorption, resulting in dehydration and acidification of the airway surface liquid (ASL). This abnormal ASL constantly hampers the mucociliary clearance mechanism (MCC) and compromises the immune defenses of the airways [6]. The formation of thick and sticky mucus creates ideal conditions for bacterial colonization [7]. The most common bacteria in CF lungs are *Pseudomonas aeruginosa*, *Staphylococcus aureus*, *Haemophilus influenzae,* and *Burkholderia cepacia* [8,9]. These bacteria easily form biofilms, which significantly increase their resistance to antibiotics. Since the eradication of bacterial biofilms is extremely difficult, chronic airway inflammation and lung damage frequently occur in CF [10].

Failure of HCO_3_^−^ secretion could be of critical importance in CF [5,11]. In normal conditions, mucus secretion requires HCO_3_^−^ to release and unfold mucin molecules by raising pH and removing Ca^2+^ [12]. In a piglet trachea model, insufficient HCO_3_^−^ secretion has been shown to cause defective mucus homeostasis [13]. A more recent study showed that CF airway abnormalities in the CFTR-knockout rat could be reversed by HCO_3_^−^ [14]. Garcia et al. reported that PGE_2_ or 5-HT stimulated mucus release was reduced by approximately one half in the absence of HCO_3_^−^ [15]. Moreover, mucus on the ileal mucosa of the CF mouse was denser and thicker than in the wild-type mouse. However, the abnormal properties of this mucus could be corrected by NaHCO_3_ supplementation [16]. Sodium bicarbonate also restores airway bacterial killing capacity in vivo [17], and we have shown that NaHCO_3_ is beneficial to ∆F508-CFTR expressing-CFBE cells [18]. The administration of NaHCO_3_ is clearly a promising therapeutic approach for treating CF lung disease.

We have recently demonstrated that adding HCO_3_^−^ inhibits the growth of planktonic CF pathogens and the formation of *P. aeruginosa* biofilms in conventional microbiological media [19]. However, a number of studies suggest that the microbiological media are designed to be convenient, rather than to resemble the complexity of the host environment [20,21]. In the CF lung, ASL has a unique composition that could influence bacterial behavior significantly [22]. We have therefore tested the effects of HCO_3_^−^ on bacterial growth and biofilm formation in an artificial sputum medium (ASM) that mimics the properties of ASL in CF patients [20,22,23]. Our data show that both bacterial growth and biofilm formation are inhibited by HCO_3_^−^ in a concentration-dependent manner.

## 2. Results

### 2.1. Sodium Bicarbonate Inhibits the Growth of S. aureus and P. aeruginosa in Artificial Sputum Medium

#### 2.1.1. Colony-Forming Unit Assays

The inclusion of 100 mM NaHCO_3_ in the ASM (pH 8.0) resulted in a significant reduction in viable cell counts for both *S. aureus* and *P. aeruginosa* after a 6 h incubation when compared to NaHCO_3_-free ASM at the same pH (Figure 1A–D). With 25 mM NaHCO_3_ in the ASM (pH 7.4) only the *P. aeruginosa* ATCC 27853 cell counts were significantly reduced (Figure 1C). Similar inhibitory effects of 100 mM NaHCO_3_ on *S. aureus* and *P. aeruginosa* growth were observed after a 17 h incubation (Figure 1E–H), whereas 25 mM NaHCO_3_ caused count reductions for *P. aeruginosa* in both the ATCC strain and the clinical isolate (Figure 1G,H).

The count reduction caused by 100 mM NaHCO_3_ after 6 h was significantly greater than that caused by 25 mM NaHCO_3_ (with the single exception of the *S. aureus* ATCC strain), suggesting a concentration-dependent inhibitory effect of HCO_3_^−^ on bacterial growth (Figure 1B–D). After the longer 17 h incubation, there were no significant differences between the inhibitory effects of 25 and 100 mM NaHCO_3_ in either species (Figure 1E–H).

To test whether these inhibitory effects could be attributed to the differences in pH of the NaHCO_3_-containing media, bacterial counts were compared in NaHCO_3_-free ASM at the same two pH values: 7.4 and 8.0. Data from *S. aureus* showed that, in the absence of NaHCO_3_, the more alkaline pH 8.0 medium did not reduce the cell counts compared to pH 7.4 (Figure 1A,B,E). Interestingly, the counts were actually increased at pH 8.0 in the *S. aureus* clinical isolate following the 17 h incubation (Figure 1F). On the other hand, the more alkaline pH slightly reduced the cell count for the *P. aeruginosa* ATCC strain after the 6 h incubation (Figure 1C), whereas no difference in effect was detected following the longer 17 h incubation (Figure 1G) or in the clinical isolate, regardless of incubation time (Figure 1D,H). Taken together, our data suggest that NaHCO_3_ has a concentration-dependent inhibitory effect on bacterial growth, which is not due to the accompanying changes in external pH.

#### 2.1.2. Flow Cytometric Assays

We used flow cytometric techniques to examine the effects of NaHCO_3_ on the membrane integrity of the *S. aureus* and *P. aeruginosa* ATCC strains grown in ASM (Figure 2). When the bacteria were treated with propanol (70% (*v*/*v*)), propidium iodide (PI) entered the cells and induced strong red signals (R-2) while the SYTO9 green signals (R-3) were invisible (Figure 2A,F), indicating that, as expected, membrane damage is linked with an increase in PI and a decrease in SYTO9 signals.

We detected changes in both the density and shape of the clusters for the SYTO9 and PI signals when NaHCO_3_ was present in the ASM (Figure 2C,E,H,J) compared to NaHCO_3_-free conditions at the same pH (Figure 2B,D,G,I). Both 25 and 100 mM NaHCO_3_ enhanced the intensity of the PI signals, indicating that NaHCO_3_ increased the bacterial membrane permeability.

Percentages of SYTO9- and PI-positive cells were compared for ASM with and without NaHCO_3_ at different pH values (Figure 3). Incubation with 100 mM NaHCO_3_ (pH 8.0) significantly reduced the percentage of SYTO9-positive cells, for both *S. aureus* and *P. aeruginosa*, when compared to NaHCO_3_-free ASM at the same pH (Figure 3A,C). Conversely, the percentage of PI-positive cells was increased in 100 mM NaHCO_3_, for both species, when compared to NaHCO_3_-free ASM at the same pH (Figure 3B,D). In addition, comparing the effects of 25 mM and 100 mM NaHCO_3_, we observed a concentration-dependent decrease in the percentage of SYTO9-positive *S. aureus* cells (Figure 3A). It is of note that in *P. aeruginosa*, 25 mM NaHCO_3_ significantly increased the percentage of PI-positive cells (Figure 3D). Furthermore, the ratio of SYTO9- to PI-positive cells remained unchanged in ASM without NaHCO_3_ when the pH values were increased from 7.4 to 8.0 (Figure 4). These data indicate that the effects of NaHCO_3_ were not attributable to the alkalinization of extracellular milieu.

### 2.2. Sodium Bicarbonate Inhibits Biofilm Formation by P. aeruginosa in ASM

We have demonstrated that biofilm formation by *P. aeruginosa* can be inhibited by both 50 and 100 mM NaHCO_3_ in conventional bouillon medium [19]. Here, we investigated whether these inhibitory effects are also observed in ASM. *P. aeruginosa* 17808 (clinical isolate) was grown in ASM for 48 h, after which biofilm formation capacity was assessed by crystal violet staining. Inclusion of both 25 and 100 mM NaHCO_3_ in the ASM inhibited biofilm formation significantly compared with the NaHCO_3_-free ASM at the same pH values (Figure 5A,B). Interestingly, the more alkaline external pH actually increased biofilm formation in NaHCO_3_-free ASM (Figure 5B, clear columns), reinforcing the conclusion that it cannot be the high pH that causes the strong inhibitory effects of 100 mM HCO_3_^-^ on bacterial growth.

## 3. Discussion

In this study, we used a customized artificial sputum medium to investigate the effects of HCO_3_^−^ on growth and biofilm formation by CF-related bacteria. We found that (i) HCO_3_^−^ inhibits the growth of both *P. aeruginosa* and *S. aureus* in a concentration-dependent manner, (ii) the inhibitory effects are probably related to bacterial membrane damage induced by HCO_3_^−^ and (iii) HCO_3_^−^ inhibits biofilm formation by *P. aeruginosa*. All of these effects were due to HCO_3_^−^
*per se* and not to changes in external pH or osmolarity.

Previous studies have pointed out the critical significance of impaired HCO_3_^−^ transport in CF pathology [5,11,13,14,15,16,17,19]. Defective HCO_3_^−^ secretion contributes to the dehydration and acidification of the ASL, impaired mucociliary clearance, and mucus accumulation, all of which provide ideal conditions for bacterial colonization [7,8]. To mimic this environment, a special medium (ASM) was developed that may help to better understand the bacterial behavior in CF airways [20,22]. The effects of each ASM component were studied on the behavior of *P. aeruginosa*. It has been shown that mucin, eukaryotic DNA, amino acids, iron chelator, lecithin, and salts were required for the formation of tight micro-colonies [22]. Thomas et al. have also demonstrated that sputum amino acid concentrations correlate with the severity of CF lung disease [24]. Most CF pathogens grown in ASM show their normal characteristics, including specific gene expression, strong micro-colony formation, metabolite utilization, surface motility, evolutionary diversity and, most importantly, rigorous antibiotic resistance, which likely reflect the real pathological situation in the lung [20,22,23,25,26,27,28,29,30]. Thus, the accumulated evidence confirms the suitability of ASM for microbiological CF research.

Our data show that 100 mM HCO_3_^−^ inhibits the growth of both *S. aureus* and *P. aeruginosa* in ASM as assessed by CFU assay. These inhibitory effects were observed following both 6- and 17-h incubations, indicating that bacterial growth was hampered both in exponential and stationary phases. These findings are in accordance with previous observations demonstrating the antimicrobial effects of NaHCO_3_ [19,31,32,33]. Corral and colleagues reported that 120 mM NaHCO_3_ reduces the CFU counts of *E. coli*, *L. plantarum*, *S. aureus* and *P. aeruginosa* 10,000-fold. This effect is more pronounced in yeasts such as *S. cerevisiae* and *H. wingei*, with 100,000-fold growth reductions induced by 60 mM NaHCO_3_ [31]. More recently, it has been demonstrated that NaHCO_3_ has antimicrobial effects on *E. coli* when applied in concentrations higher than 25 mM [32]. Our previous work has also shown that the growth of common CF bacteria is inhibited in brain-heart infusion (BHI) media containing NaHCO_3_ [19].

Flow cytometry can be used in combination with nucleic acid double-staining (NADS) for viability determination based on bacterial membrane integrity [34]. Our data shows that ASM containing 100 mM NaHCO_3_ significantly increases the percentage of damaged cells, while alkalinization of the medium *per se* (from pH 7.4 to 8.0) has no effect on *S. aureus* (a Gram-positive bacterium). Repeating these experiments with *P. aeruginosa* was not at first successful because the percentage of SYTO9-positive cells remained very low even under control conditions. Other studies have reported the same phenomenon when flow cytometry and live/dead staining is used with Gram-negative bacteria [35,36,37]. The outer membrane of these bacteria constitutes a significant barrier to SYTO9 permeation. However, Berney and colleagues have shown that an appropriate dose of UVA light or EDTA (5 mM) can weaken the outer membrane and allow SYTO9 uptake [36]. Indeed, when treating the cells with EDTA (5 mM), we were able to demonstrate that 100 mM HCO_3_^−^ has inhibitory effects on *P. aeruginosa* similar to those we observed on *S. aureus*. These results confirm the antimicrobial properties of NaHCO_3_ and suggest that the effects we observed were mainly due to bacterial membrane damage.

The molecular mechanisms behind the antibacterial effects of HCO_3_^−^ are still unclear. We hypothesize that they might be due to the ability of HCO_3_^−^ to chelate divalent cations (Ca^2+^ and Mg^2+^). Lowering external divalent cation concentrations may increase bacterial membrane fragility and subsequently decrease viability. Indeed, divalent cations are known to maintain bacterial outer membrane integrity [38,39]. Not surprisingly, therefore, EDTA has inhibitory effects on the growth and biofilm formation of *S. aureus* [40,41,42]. In our experiments, EDTA enhanced SYTO9 entry into *P. aeruginosa,* which is probably due to outer membrane destabilization and increased permeability [43]. It has also been suggested that HCO_3_^−^ may dissipate the proton gradient across the bacterial membrane, which could also decrease bacterial viability [44]. Although our data suggest that increased osmolarity did not contribute to the inhibitory effects of HCO_3_^−^, we do not exclude the possibility that NaHCO_3_^−^ containing hypertonic solutions might have antibacterial effects, as has been observed with hypertonic saline [45].

The inhibitory effects of 100 mM HCO_3_^−^ exceeded those of 25 mM HCO_3_^−^ after 6 h of incubation, but not after 17 h, where both concentrations were equally effective. We speculate that during the longer incubations, some loss of HCO_3_^−^ may have occurred. Therefore, coupled with the fact that HCO_3_^−^ is bacteriostatic rather than bactericidal [19], bacteria may recover from the temporary growth inhibition. Our CFU assays and flow cytometry data also showed that the lower concentration of HCO_3_^−^ (25 mM) reduced the growth of *P. aeruginosa* but not *S. aureus*, suggesting that bacterial species may have differing susceptibilities to HCO_3_^−^.

In the CF lung, bacterial biofilm formation presents major challenges for clinicians. Since we have previously shown that HCO_3_^−^ inhibits *P. aeruginosa* biofilm formation in bouillon supplemented with 2% glucose, we here investigated its effects in ASM. Under these conditions, both 25 and 100 mM NaHCO_3_ significantly inhibited biofilm formation, thus strengthening the clinical relevance of our previous observations [19]. Interestingly, we detected a significant increase in *P. aeruginosa* biofilm formation in alkaline (pH 8.0) NaHCO_3_-free ASM. This observation is consistent with previous findings showing that pH influences biofilm formation [46,47]. However, despite the stimulatory effect of pH 8.0 on *P. aeruginosa* in the absence of HCO_3_^−^, the ASM containing 100 mM NaHCO_3_, also pH 8.0, caused a marked inhibition of biofilm growth. This indicates that the more powerful antibacterial effect of 100 mM HCO_3_^−^ can override any effects of the accompanying pH change.

In conclusion, we have demonstrated that HCO_3_^−^, and not raised pH or osmolarity inhibits both the growth and biofilm formation of prevalent CF bacteria grown in an artificial sputum medium whose composition resembles viscous CF mucus. It seems likely that HCO_3_^−^ increases the permeability of the bacterial membrane, thus reducing cell viability. Bicarbonate should, therefore, be considered a potentially valuable therapeutic agent in CF and other chronic airway diseases involving bacterial infections.

## 4. Materials and Methods

### 4.1. Artificial Sputum Medium and Growth Conditions

ASM was prepared from sterile stock solutions, with the final concentrations shown in Table 1. The mucin stock solution (5% (*w*/*v*)) was prepared with deionized water sterilized by autoclaving at 121 °C for 30 min. Salmon testis DNA and ferritin were dissolved in sterile deionized water. All other stock solutions were prepared and sterilized using 0.22 µm-pore syringe filters.

Artificial sputum media were prepared with and without HCO_3_^−^ (Table 2). Please note that the ASM used in the flow cytometry experiments did not contain DNA because SYTO9 and PI, being nucleic acid stains, would bind to extracellular DNA.

The pH of the NaHCO_3_-free ASM was adjusted by mixing appropriate volumes of the HEPES acid and Na-HEPES stock solutions. For the NaHCO_3_-containing ASMs, the components of NaCl were reduced to maintain isosmotic conditions. Thus the NaCl concentration was reduced to 75 mM in the ASM containing 25 mM NaHCO_3_, and to zero in the ASM containing 100 mM NaHCO_3_. Both of the NaHCO_3_-containing ASMs were incubated in 5% CO_2_ giving pH 7.4 and 8.0, respectively. Since NaHCO_3_ is heat-sensitive, 0.22 µm-pore syringe filters were used to sterilize the NaHCO_3_ stock solution. Furthermore, NaHCO_3_ was added to each ASM immediately prior to bacterial inoculation.

### 4.2. Bacterial Strains

*Staphylococcus aureus* (ATCC^®^ 29213™) and *Pseudomonas aeruginosa* (ATCC^®^ 27853™) were used in this study. Colony-forming unit (CFU) assays were also carried out with clinical isolates of the same bacterial species (*S. aureus* SA-113 and *P. aeruginosa* 17808). The pre-culture of each bacterium was prepared for each individual experiment from the same stock culture stored at −80 °C. Bacteria were plated onto simple agar plates and incubated overnight. Single colonies were then picked to inoculate into 15-mL tubes containing 5 mL BHI broth (Mast Group Ltd., Merseyside, UK) and cultured overnight at 37 °C. The density of the cultures was adjusted with a VITEK Densichek apparatus (Biomérieux, Marcy l’ Étoile, France) directly before using them for the experiments.

### 4.3. Growth Experiments

#### 4.3.1. Colony-Forming Unit Assay

Overnight cultures of the bacteria were adjusted to 3.0 McFarland (approximately 9.0 × 10^8^ cells/mL) and subcultured at a 1:50 dilution into ASM and mixed gently. 200 µL aliquots of each suspension were dispensed into 96-well plates in triplicate and incubated at 37 °C in ambient air (ASM without NaHCO_3_) or 5% CO_2_ (ASM with either 25 or 100 mM NaHCO_3_) (Table 2). After 6 or 17 h incubation, 30 µL of the bacterial culture was taken and serially diluted over a range of dilution factors from 10^−1^ to 10^−9^. Then 10 µL aliquots of each dilution were plated onto simple agar plates, which were incubated overnight at 37 °C. The colonies on each plate were counted using ImageJ software (NIH, Bethesda, MD, USA). Only plates that showed between 25 and 250 colonies were selected, and the colony densities (CFU/mL) were calculated using the following equation:CFU/mL = (number of counts on the plate)/(0.01 × dilution factor)(1)

Results in CFU/mL were then converted to a logarithmic scale (log CFU/mL). In each condition, three independent experiments were carried out (*n* = 3). All data were pooled (totaling 9 replicates per treatment group, except for *S. aureus* ATCC at 6 h having only 3 replicates). The mean values of log CFU/mL in each condition were compared as designated.

#### 4.3.2. Flow Cytometry

##### Bacterial Cultures and Sample Preparation before Staining

Overnight cultures were adjusted to 3.0 McFarland and subcultured at a 1:50 dilution into DNA-free ASM. 200 µL aliquots of each suspension were dispensed into sterile 1.5-mL tubes in triplicate and subsequently incubated for 17 h at 37 °C in ambient air or 5% CO_2_. After incubation, 0.85% NaCl solution (1 mL) was added to each tube, which was then centrifuged at 12,000 rpm for 2 min at room temperature (RT). The pellet was re-suspended in 1 mL 0.85% NaCl solution and incubated for 10 min at RT. This step was repeated twice to remove excess ASM. Each bacterial suspension was then adjusted with 0.85% NaCl solution to 0.5 McFarland (approx. 1.5 × 10^8^ cells/mL). In experiments with *P. aeruginosa*, EDTA (5 mM) was added to the saline solution to disrupt the outer membranes of the bacteria and to facilitate penetration of the dye [36]. 

##### Staining Procedure

Bacterial suspensions were stained with the LIVE/DEAD BacLight Bacteria Viability Kit (L7007, Invitrogen, Waltham, MA, USA). The BacLight consists of SYTO9, a membrane-permeant dye penetrating all cells, and PI, which is cell-impermeant and only enters damaged or dead cells. The staining reagent was prepared according to the manufacturer’s instructions. Briefly, component A (1.67 mM SYTO9/1.67 mM PI) and component B (1.67 mM SYTO9/18.3 mM PI) were mixed 1:1 in a microtube. Five microliters of the mixture was added to 1 mL of each bacterial suspension (5 µL/mL final concentration). The suspensions were subsequently mixed thoroughly and incubated in the dark for 25 min before measurement at RT. Microbeads (100 µL) (Invitrogen, USA) were added to the suspensions for cell quantification. Samples containing ASM without bacteria were prepared and stained to verify background noise. The autofluorescence of the bacteria was assessed using unstained cells, and positive controls were generated by pre-treating the cells with propanol (70% (*v*/*v*)) to cause membrane damage, maximizing PI penetration. Therefore, the membrane-damaged or dead cells were simply detected with the high intensity of PI.

##### Flow Cytometric Measurement

Flow cytometry was carried out using a BD FACSCalibur system (Becton Dickinson, San Jose, CA, USA) equipped with a 635-nm red diode laser and a 15-mW 488-nm air-cooled argon solid-state laser. Forward scatter (FSC) and side scatter (SSC) were collected from the 488 nm excitation. SSC was set as a discriminator to reduce electronic background noise during the analysis. 

The instrument settings were defined by the Megamix-Plus SSC beads (Biocytex, Marseille, France) and were optimized with 1 µm Silica Beads Fluo-Green Green (Kisker Biotech GmbH & Co., Steinfurt, Germany). Stained bacteria were excited by the 488-nm laser, and the fluorescence was collected through a 530/30-nm bandpass filter (SYTO9) and a 670-nm long-pass filter (PI). All signals were amplified logarithmically (four decades). The sampling rate was adjusted to less than 1000 particles/s. Each measurement lasted 1 min. Sterile PBS was applied as a sheath fluid. Data were acquired with BD CellQuest Pro software (Becton Dickinson, San Jose, CA, USA). Stained cell suspensions were analyzed immediately after dye incubation.

##### Flow Cytometry Data Analysis

Data gating and analysis were performed using Flowing software version 2.5.1 (Turku Centre for Biotechnology, Turku, Finland, released 4.11.2013). Dot plots of detected signals from each sample were analyzed based on the FSC, SSC, green (FL1), and red (FL3) fluorescence intensities (Figure 6). 

A standardized bacterial gate (R-0) was set on the FSC-SSC dot plot, based on the Megamix-SSC boundary, to select the bacterial population (purple, Figure 6A). This bacterial R-0 gate was then saved and applied to other samples. Next, the unstained, propanol-treated, and untreated (NaHCO_3_-free) samples were analyzed to determine the regions of autofluorescence, dead and living cells, respectively (Figure 6B–D). Briefly, signals detected from the unstained sample were identified first and attributed to autofluorescence and background noise. They were subsequently gated in R-1 (grey, Figure 6B) to discriminate them from bacterial signals. Next, the positive control and untreated samples were analyzed. Signals with high intensities in FL1 and FL3 (SYTO9- and PI-positive signals, respectively) were selected and attributed to bacterial cells (Figure 6C,D). For the determination of bacterial viability, manually set gates were applied based on the positive control and untreated samples. Since the positive control samples were treated 70% propanol in order to kill the bacteria, the signals detected from this sample, exhibiting a high intensity in FL3, were attributed to membrane-damaged or dead cells, and gated in the R-2 region (red, Figure 6C). This R-2 gate was next applied to the untreated samples, where signals outside the R-2 gate, which exhibited a high intensity in FL1, were gated in the R-3 region (green, Figure 6D) and attributed to membrane-intact or healthy cells. The presets of these R-regions were saved as a template and then applied to the other samples using the automated folder runner function to obtain the data from each sample. 

Frequencies of the signals inside the R-2 and R-3 gates from each condition were quantified using the statistic function of the software and presented as percentages of the total population in R-0. Three (*P. aeruginosa*) and four (*S. aureus*) independent experiments were performed. All data were pooled (totaling 9–12 replicates per treatment group). Means of the percentages of SYTO9- and PI-positive signals, and the ratio of SYTO9- to PI-positive signals were compared in ASM with and without NaHCO_3_.

### 4.4. Biofilm Crystal Violet Assay

Bacterial suspensions of *P. aeruginosa* 17808 (clinical isolate) were prepared following the same protocol as described for the CFU assays. 200 µL of each suspension was dispensed into 96-well plates in five duplicates. Bacteria were incubated in ambient air (ASM without NaHCO_3_) or 5% CO_2_ (ASM with either 25 or 100 mM NaHCO_3_) for 48 h at 37 °C. After incubation, unattached bacteria were removed by rigorous washing with 200 µL of 1× PBS three times. Bacteria attached to the wells were air-dried and subsequently stained with 125 µL crystal violet solution (0.1%) for 10 min. Excess crystal violet was removed by rinsing the plates several times in tap water. The plates were then air-dried. Crystal violet stain was solubilized in 30% acetic acid (200 µL/well) for 10 min. From each well, 125 µL of this solution was taken and transferred to a new flat-bottom 96-well plate (Figure 5A). Optical density (OD) was measured at 595 nm in a PR2100 microplate reader. The average OD from the control ASM wells without bacteria was subtracted from the ODs measured in wells with bacteria. Three independent experiments were performed (*n* = 3). All data were pooled (totaling 15 replicates per treatment group). Means of OD in each condition were compared in the ASM with and without NaHCO_3_.

### 4.5. Statistical Analysis

The statistical analysis was performed on the pooled data of each experimental group, except for *S. aureus* ATCC 29213 CFU assay at 6 h, by using GraphPad Prism version 8.0.0 (GraphPad Software, Inc., San Diego, CA, USA). Pooled data were normally distributed (Shapiro–Wilk test) and presented as means ± SD. The means were compared using one-way ANOVA, followed by Tukey’s post-hoc multiple comparison test. The un-pooled data were analyzed by using Kruskal–Wallis test, followed by Dunn’s post-hoc multiple comparison test. Changes were considered statistically significant if *p* < 0.05.

## Figures and Tables

**Figure 1 ijms-21-08614-f001:**
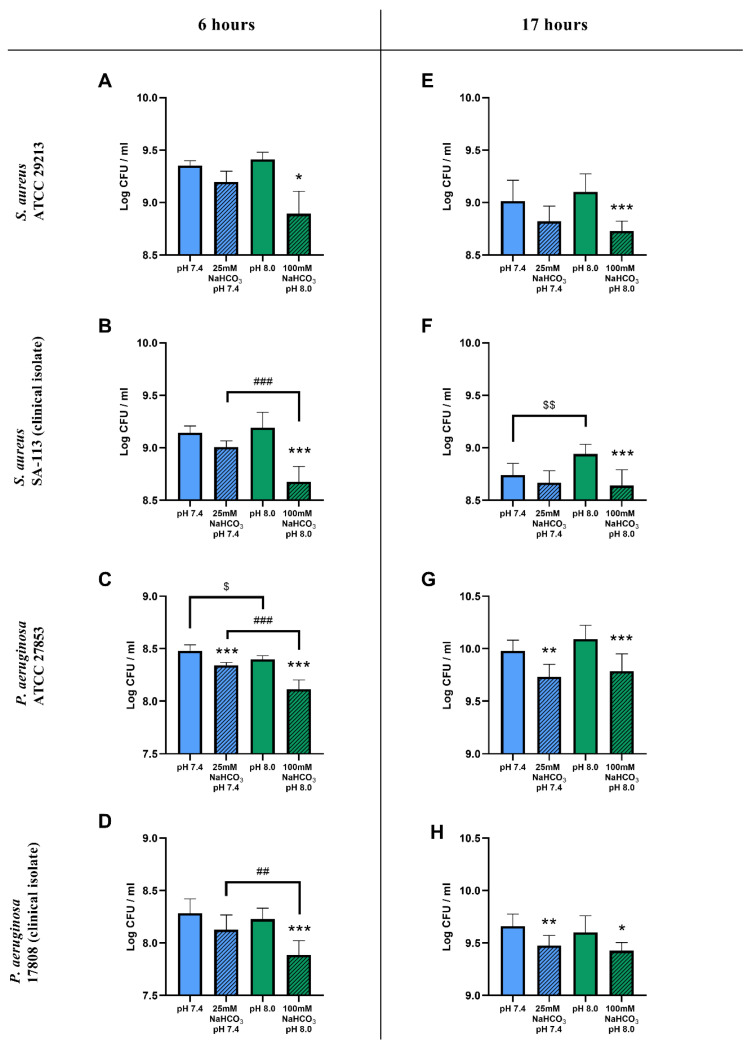
CFU assay of cystic fibrosis bacteria: *S. aureus* ATCC 29213 (**A**,**E**), *S. aureus* SA-113 (**B**,**F**), *P. aeruginosa* ATCC 27853 (**C**,**G**), and *P. aeruginosa* 17808 (**D**,**H**) grown in different ASM conditions for 6 (**A**–**D**) and 17 h (**E**–**H**) in ambient air or 5% CO_2_. Values are presented as means of log CFU/mL ± SD. The experiment was repeated three times. All data were pooled, totaling 9 replicates per treatment group, except for *S. aureus* ATCC at 6 h having only 3 replicates (**A**). Statistical analysis: one-way ANOVA followed by Tukey’s post-hoc multiple comparison test (**B**–**H**) or Kruskal–Wallis test followed by Dunn’s post-hoc multiple comparison test (**A**). * = *p* < 0.05, ** = *p* < 0.01, *** = *p* < 0.001 when comparing ASM with and without NaHCO_3_ at the same pH (same-colored columns); ## = *p* < 0.01, ### = *p* < 0.001 when comparing the two NaHCO_3_ concentrations (25 vs. 100 mM) (shaded columns); $ = *p* < 0.05, $$ = *p* < 0.01 when comparing NaHCO_3_-free ASM at pH 7.4 and 8.0 (clear columns).

**Figure 2 ijms-21-08614-f002:**
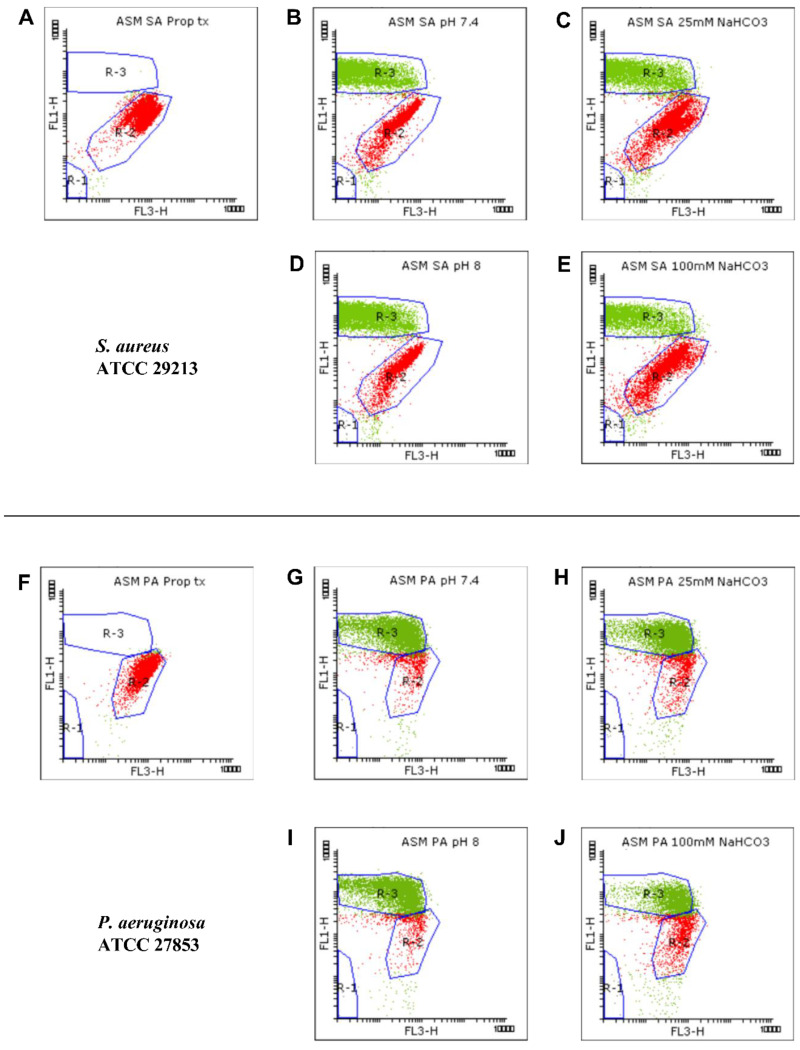
Viability staining for analyzing the growth of *S. aureus* (SA) ATCC 29213 (**A**–**E**) and *P. aeruginosa* (PA) ATCC 27853 (**F**–**J**). Bacteria were cultured in ASM and treated with 70% propanol for membrane permeabilization (**A**,**F**), in NaHCO_3_-free ASM (pH 7.4) (**B**,**G**), in ASM containing 25 mM NaHCO_3_ (pH 7.4) (**C**,**H**), in NaHCO_3_-free ASM (pH 8.0) (**D**,**I**), and in ASM containing 100 mM NaHCO_3_ (pH 8.0) (**E**,**J**). Bacterial signals from each condition are plotted as dot plots (FL1 vs. FL3). SYTO9-positive (green), PI-positive (red), and autofluorescence signals were gated in R-3, R-2, and R-1, respectively.

**Figure 3 ijms-21-08614-f003:**
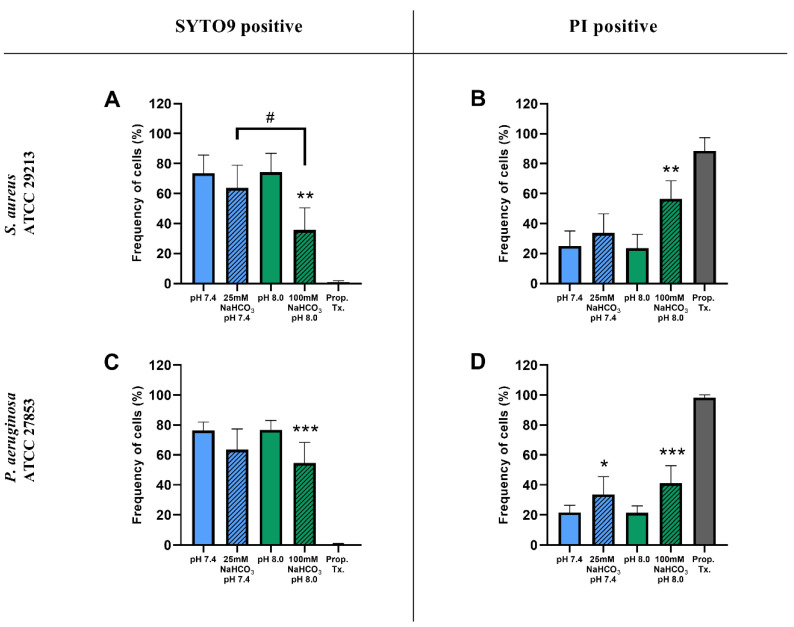
Percentage of SYTO9- and PI-positive signals of *S. aureus* ATCC 29213 (**A**,**B**) and *P. aeruginosa* ATCC 27853 (**C**,**D**) cells grown in different ASM after 17 h of incubation. Values are presented as percentage means ± SD. The experiment was repeated three (*P. aeruginosa*) or four times (*S. aureus*), totaling 9–12 replicates per treatment group. Statistical analysis: one-way ANOVA followed by Tukey’s post-hoc multiple comparison test. * = *p* < 0.05, ** = *p* < 0.01, *** = *p* < 0.001 when comparing ASM with and without NaHCO_3_ at the same pH (same-colored columns); # = *p* < 0.05 when comparing the two NaHCO_3_ concentrations (25 vs. 100 mM) (shaded columns).

**Figure 4 ijms-21-08614-f004:**
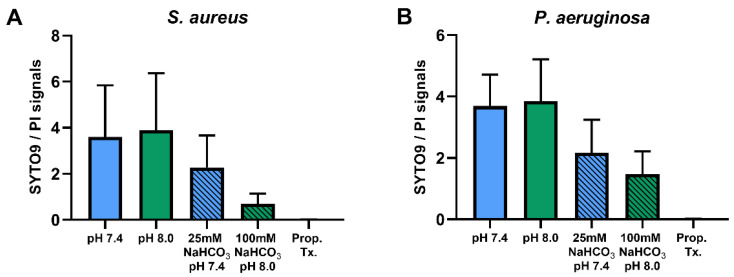
Ratios of SYTO9- to PI-positive signals of *S. aureus* ATCC 29213 (*n* = 4) (**A**) and *P. aeruginosa* ATCC 27853 (*n* = 3) (**B**) in different ASM media. Values are presented as means ± SD.

**Figure 5 ijms-21-08614-f005:**
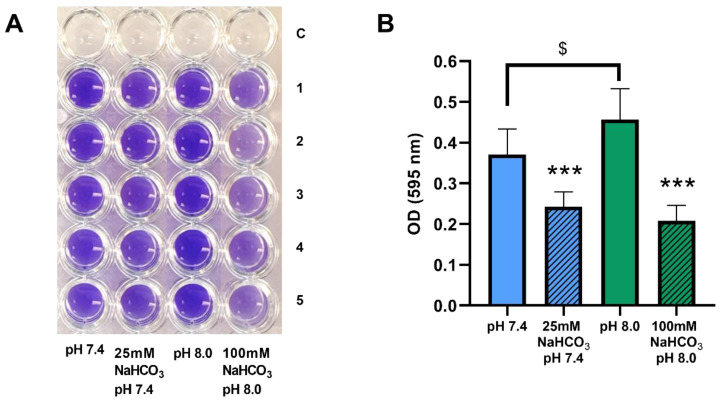
Biofilm formation by *P. aeruginosa* 17808 (clinical isolate) grown in different ASM. (**A**) crystal violet staining after 48 h of incubation. (**B**) statistical analysis of biofilm formation. Values are presented as means of optical density (OD) ± SD. The experiment was repeated three times (*n* = 3, 15 replicates per treatment group). Statistical analysis: one-way ANOVA followed by Tukey’s post-hoc multiple comparison test. *** = *p* <0.001 when comparing ASM with and without NaHCO_3_ at the same pH (same-colored columns); $ = *p* <0.05 when comparing the NaHCO_3_-free ASM at pH 7.4 and 8.0 (clear columns).

**Figure 6 ijms-21-08614-f006:**
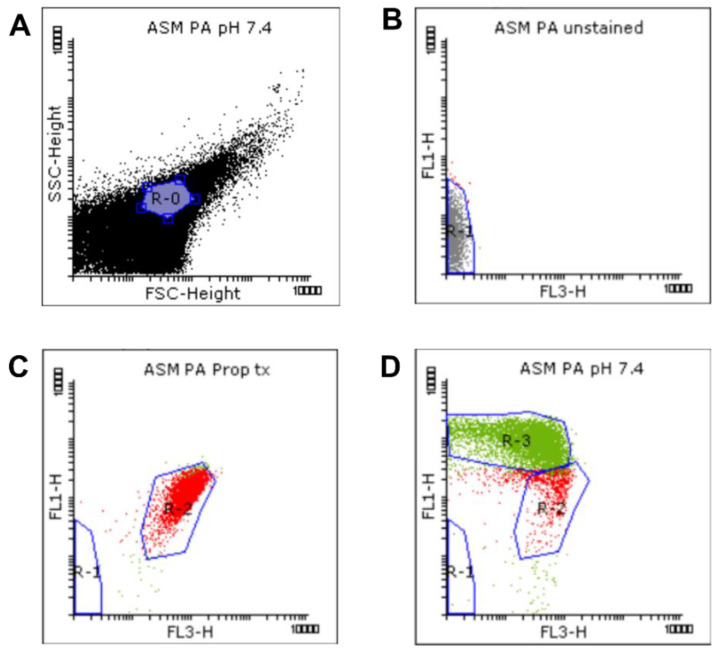
Gating strategy for the flow cytometric measurement. Viability was determined by membrane integrity analysis using the LIVE/DEAD BacLight Bacteria Viability Kit. We defined a standardized bacterial gate (R-0) (**A**) on the FSC-SSC dot plot, on the basis of the Megamix-SSC boundary. The bacterial signals inside the R-0 gate of the unstained (**B**), propanol-treated (**C**), and untreated (NaHCO_3_-free) samples (**D**) were analyzed based on the FL1 vs. FL3 fluorescence. Autofluorescence signals were gated in R-1. Signals with high FL3 intensity were gated in R-2. Signals with high FL1 intensity were gated in R-3.

**Table 1 ijms-21-08614-t001:** ASM components and their final concentrations.

Name	Stock Concentration	Final Concentration
Mucin from porcine stomach	5% (*w*/*v*)	2% (*w*/*v*)
DNA sodium salt from salmon testes	14 mg/mL	1.4 mg/mL
Casein hydrolysate	20 mg/mL	5 mg/mL
Egg yolk emulsion	1×	0.005×
Ferritin	1 mg/mL	0.003 mg/mL
NaCl *	2 M	100, 75, or 0 mM
NaHCO_3_ *	1 M	0, 25, or 100 mM
KCl	2 M	30 mM
Glucose	2 M	11 mM
HEPES acid	1 M	50 mM
HEPES Na salt	1 M	

ASM recipe is modified from Sriramulu et al. (2005) and Quinn et al. (2015) [22,29]. * Concentration of NaCl and NaHCO_3_ are varied depending on the ASM conditions (Table 2).

**Table 2 ijms-21-08614-t002:** ASM conditions and modifications.

Test	Conditions	pH	DNA	NaCl (mM)	NaHCO_3_ (mM)	Atmospheric Condition
CFU and biofilm experiments	(1) NaHCO_3_-free ASM	7.4	Present	100	-	Ambient air
	(2) NaHCO_3_-free ASM	8.0	Present	100	-	Ambient air
	(3) 25 mM NaHCO_3_-ASM	7.4	Present	75	25	5% CO_2_
	(4) 100 mM NaHCO_3_-ASM	8.0	Present	-	100	5% CO_2_
Flow cytometry	(1) NaHCO_3_-free ASM	7.4	Absent	100	-	Ambient air
	(2) NaHCO_3_-free ASM	8.0	Absent	100	-	Ambient air
	(3) 25 mM NaHCO_3_-ASM	7.4	Absent	75	25	5% CO_2_
	(4) 100 mM NaHCO_3_-ASM	8.0	Absent	-	100	5% CO_2_

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
