# Peer review of "Antibacterial Effects of Bicarbonate in Media Modified to Mimic Cystic Fibrosis Sputum"

_ijms, 2020, doi:10.3390/ijms21228614_

Round 1
Reviewer 1 Report
Review of manuscript by Jaikumpun et al.
Antibacterial Effects of Bicarbonate in Media Modified to Mimic Cystic Fibrosis Sputum.
Major results: The authors repeated a previous study aimed at determining the impact of bicarbonate ion concentration on antibacterial activity of CF sputum. The previous study examined the impact of manipulations of bicarbonate concentration on a “broth” version of sputum. This time, the effect of increasing bicarbonate concentrations was tested in a more authentic mimic of sputum called ASM (airway surface medium). The authors have already published the composition of ASM.
The authors, studied two clinical isolates of P.aeruginosa and two isolates of S.aureus with respect to changes in their viability with increasing bicarbonate concentrations in ASM. The authors show that bicarbonate (at 100 mM and 25 mM for certain isolates) decreased viable planktonic bacterial count and biofim formation and this effect was independent of alkalinization and osmolarity.
Critique: The results were mostly convincing- but they seemed somewhat predictable based on the groups previous findings. Given the effort taken by this group to recapitulate the properties of CF mucus, it will be important to determine if this composition is important. For example, what is the difference between the findings reported for the first and second mucus model systems tested. What components in the two mucus models could potentially modify the bicarbonate activity? The role of components that are unique in ASM and potentially important for regulating bicarbonate concentration should be tested.
Have the authors considered conducting the same studies using patient specific sputum? Do they expect to see differential effects of bicarbonate ion on bacteria depending on unique characteristic and sputum from different donors?
Overall, the results are convincing but their novelty is not clear to this reviewer.
Reviewer 2 Report
The article by Jaikumpun et al. is a good piece of work, and extremely interesting in the field of cystic fibrosis. Using an in vitro method, with a model of airways secretions, they demonstrate the important role of the presence of bicarbonate, beyond the effect on pH, in bacterial growth in the respiratory tract. This information constitutes a further guide in the design of cystic fibrosis therapy, a condition in which bicarbonate secretion is compromised.
However, from a formal point of view, the article deserves a review. Most of the experiments have been done with only 3 replicates. This small number of replicates makes it impossible to determine whether the data are normally distributed, and consequently, their presentation as averages and standard deviations is invalid. On the other hand, the experimental groups are compared with a poor defined "one-way ANOVA and multiple comparison test". I should point out that the test of variances (ANOVA) is designed for normally distributed samples. In addition, there are a dozen different post-hoc multiple comparison tests; it would be important to specify which one has been used.
Minor issues.
In lines 51-62 it is emphasized that the presence of bicarbonate is necessary for the secretion of mucus. In reality, Garcia et al. Affirm that bicarbonate is essential for post-secretory mucus modification, and that bicarbonate efflux is necessary to maintain optimal conditions of secretion -a fluid mucus in the ASL-, and not that it conditions secretion itself.
The experiments have been done with 0, 25 and 100 mM of bicarbonate. The concentration of bicarbonate in ASL is between 10 and 25 mM, while in CF patients it is between 5 and 8 mM. Values of 0 and 100 mM bicarbonate do not make much sense in the description of CF. Although experimentally they make sense, it would be necessary to justify the choice of these concentrations in terms of the pathophysiology of CF.
Round 2
Reviewer 1 Report
I thank the authors for their response concerning the novelty of the work. They make it clear that they have established a novel medium, that better models sputum than the "broth" used in previous work. And they show convincing that this model sputum supports previous work showing the important role of bicarbonate ion in reducing viability of planktonic bacteria and biofilm growth.
However, i still maintain that the significance of the studies would be enhanced if the role of different components of the novel medium in regulating bacterial viability were defined.
Author Response
Thank you very much for your comments. We agree that it would be interesting to investigate the role of different components of the ASM in regulation of bacterial viability. However, we think that this could be the aim of future studies.
Nonetheless, we have included new sentences to the Discussion (highlighted in blue, lines 177-181).
“The effects of each ASM component were studied on behavior of P. aeruginosa. It has been shown that mucin, eukaryotic DNA, amino acids, iron chelator, lecithin, and salts were required for the formation of tight micro-colonies [Sriramulu et al., 2005]. Thomas et al. have also demonstrated that sputum amino acid concentrations correlate with severity of CF lung disease [Thomas et al., 2000].”
We have also added a new reference (Thomas et al. 2000) (lines 457-459).
Round 3
Reviewer 1 Report
The authors have addressed my concerns